# Depression Severity among a Sample of LGBTQ+ Individuals during the COVID-19 Pandemic

**Marybec Griffin** [1,2,*], **Jessica Jaiswal** [2,3,4], **Tess Olsson** [2], **Jesse Gui** [2], **Christopher B. Stults** [2,5] and **Perry N. Halkitis** [1,2,6]

1   Department of Health Behavior, Society & Policy, Rutgers School of Public Health, Rutgers University, New Brunswick, NJ 08901, USA; perry.halkitis@rutgers.edu
2   Center for Health, Identity, Behavior & Prevention Studies, Rutgers University, New Brunswick, NJ 08901, USA; jessicajaiswal@uabmc.edu (J.J.); tso21@sph.rutgers.edu (T.O.); zexigui@gmail.com (J.G.); christopher.stults@baruch.cuny.edu (C.B.S.)
3   Department of Family and Community Medicine, Heersink School of Medicine, University of Alabama at Birmingham, Birmingham, AL 35294, USA
4   Center for Interdisciplinary Research on AIDS, Yale University, New Haven, CT 06520, USA
5   Psychology Department, Baruch College, City University of New York, New York, NY 10017, USA
6   Department of Biostatistics & Social and Behavioral Health Sciences, Rutgers School of Public Health, Rutgers University, New Brunswick, NJ 08901, USA
*   Correspondence: mcg.197@sph.rutgers.edu

**Abstract:** Background: The global pandemic of coronavirus disease 2019 (COVID-19) has led to immense impacts on global community health, the public perception of healthcare, and attitudes surrounding mental health during widespread quarantine. Methods: This analysis examines the rates of depressive symptomology among a sample of LGBTQ+-identifying individuals in the United States ($n$ = 1090). The variables examined included socio-demographic factors, the use of mental health medication, access to mental health medication, and experiences of depression symptomology. Results: The findings indicate that depressive symptoms were less severe for older adults, as they reported higher levels of minimal to moderately severe depressive symptoms. Participants who were not working and those who were using substances were less likely to report depressive symptoms. Participants who were employed full-time reported higher levels of depression compared to those who were unemployed. Conclusions: Understanding the mental health of marginalized populations such as the LGBTQ+ community is critical to providing more nuanced preventative healthcare for unique populations, as members of the LGBTQ+ community are non-monolithic and require more personalized approaches to their healthcare needs.

**Keywords:** COVID-19; LGBTQ+; depression; employment status; substance use; multivariable logistic regression; pandemic impact on mental health

## 1. Introduction

The COVID-19 pandemic impacted population-level mental health in an unprecedented way. Rates of depression, anxiety, suicidality, and substance use in the general population increased as a result of the onset of the widespread viral transmission and resulting pandemic [1–5]. Efforts to contain and curtail the spread of COVID-19 resulted in stay-at-home orders and virtual work policies. It is well documented that certain public health policies such as stay-at-home orders resulted in unintended consequences, such as an increase in intimate partner violence [6], unemployment [7], and financial instability, as well as decreased sexual activity [8] and decreased adherence to medications [9]. Each of these are factors that are associated with an increase in depression and suicidal thoughts and behavior [10–12]. The long-term effects of this sudden increase in mental health burden have yet to be fully elucidated.

While extant research has extensively examined the impact of the COVID-19 pandemic on the general population's mental health, there is a dearth of research exploring the impacts of the COVID-19 pandemic on LGBTQ+ populations. Even prior to the pandemic, LGBTQ+ and other sexual- and gender-minority individuals were more likely to experience poorer physical, emotional, and mental health outcomes compared to their cisgender and heterosexual counterparts, such as elevated rates of depression, anxiety, suicidal ideation, and substance use [6,8,9,13–15]. These mental health disparities may be attributed to a variety of factors, such as discrimination and victimization based on sexual orientation and gender identity [16–18], financial disparities [19,20], and a lack of access to physical and mental health services [21–23].

It is likely that mental health disparities that were present before the COVID-19 pandemic were exacerbated due to the pandemic and resulted in poorer mental health outcomes during the pandemic for the LGBTQ+ population [19,20,24]. The extant research examining the experiences of LGBTQ+ populations holds that gender- and sexual-minority people experienced worse overall physical, financial, and mental health outcomes due to the COVID-19 pandemic compared to their cisgender and heterosexual counterparts [25,26]. The handful of studies exploring the effects of the COVID-19 pandemic on the mental health of LGBTQ+ populations found an increase in psychological distress, anxiety, depression, and alcohol use coinciding with the onset of the COVID-19 pandemic, as well as higher rates of generalized distress among sexual-minority people compared to their heterosexual counterparts [27–31].

Higher levels of stress and depression during the pandemic may be explained by experiences of discrimination related to sexual orientation [6,32–34], as well as an increase in the frequency of intimate partner violence [6,35]. Furthermore, social isolation, loneliness, and a lack of access to community groups may have more severely impacted the mental health of LGBTQ+ people due to this population's tendency to rely more on chosen family for support [31]. For example, LGBTQ+ young people who were forced to cohabitate with unsupportive family members during stay-at-home orders experienced heightened psychological distress [36,37].

Forced cohabitation with unsupportive family members is likely a function of age and employment status. Studies have shown that LGBTQ+ adolescents and young adults who were unable to return to college and were forced to live with unsupportive family members had higher rates of mental distress, including depression and suicidal ideation [26,38]. Similarly, employment loss, a well-documented phenomenon during the height of the COVID-19 pandemic [7], also led to changes in housing status [39,40], resulting in many LGBTQ+ individuals cohabitating with unsupportive family members [20,37,39]. Additionally, employment loss has a demonstrable negative on mental health, including depression and loss of identity [7,41]. Given these two factors, many individuals reported increased substance use during the height of the COVID-19 pandemic [36,37]. The reasons for substance use vary between individuals, but for many LGBTQ+ people, substance use during the COVID-19 pandemic provided a means of escape from daily stressors, serving as both a means to relax and a way to perform self-care [36,37].

This study was conducted to describe the impact of the COVID-19 pandemic on the mental health burden of LGBTQ+ adults living in the United States by measuring depressive symptoms and depression severity and to fill a gap in research concerning the mental health of LGBTQ+ adults during the COVID-19 pandemic. A better understanding of the mental health implications of the pandemic on LGBTQ+ adults is vital to formulating strategies to address the ongoing mental health impact of the COVID-19 pandemic on this population.

## 2. Materials and Methods

### 2.1. Study Design

This cross-sectional study employed an internet-based recruitment plan. All data were collected between May and July 2020. This study sought to understand the impact

of the COVID-19 pandemic on the lives of LGBTQ+ people living in the United States. The eligibility criteria for participation were as follows: participants must (1) be at least 18 years old, (2) self-identify as a member of the LGBTQ+ community, and (3) reside in the United States or the territories of the United States. The survey included measures that assessed socio-demographic factors, mental health, medication and healthcare access, sexual behaviors, substance use, and intimate partner violence. The Rutgers University Institutional Review Board approved the study protocol #Pro2020000920.

Participants were recruited via various listservs designed for LGBTQ+ individuals, professional organizations (e.g., the American Public Health Association, the American Psychological Association), and institutional and personal social media accounts (e.g., LinkedIn, Twitter, Instagram, Facebook). The recruitment materials indicated that participants would be entered into a raffle for a chance to win one of ten USD 100 gift cards if they completed the optional unlinked survey, where they were asked to provide their email addresses. Additional information about the recruitment design has been published elsewhere [42]. Interested individuals completed a screener questionnaire to determine eligibility. If the individual met the inclusion criteria, they were then asked to provide tacit consent prior to enrollment into the study. Participants first completed the COVID-19 survey and were given the ability to opt in to the incentive survey for a chance to win one of the ten USD 100 gift cards. Participants opting into the incentive survey were asked to provide their email address; no other identifying information was collected. Participants were able to complete the COVID-19 survey without completing the incentive survey.

### 2.2. Sample

The sample included 1090 participants who met the inclusion criteria. Participants provided answers to the following variables included in this analysis: age, race and ethnicity, gender identity, sexual orientation, relationship status, employment status, insurance status, lifetime use of mental health medications, and access to mental health medications. Participants also completed the Patient Health Questionnaire-9 (PHQ-9) to assess their degree of depression severity.

### 2.3. Procedures

To recruit participants, research staff members compiled a list of LGBTQ+, professional, and other community listservs that would allow survey recruitment posts. Additionally, research staff also compiled a list of institutional and personal social media accounts that would post recruitment materials. Research staff sent requests to each listserv twice to post recruitment materials. Institutional social media accounts also were asked to make two different recruitment posts during the period of survey administration. Individuals posting to personal social media accounts were allowed to post as frequently as they wanted. Research staff were not able to track the numbers of forwards, shares, retweets, and reposts from all recruitment efforts. The COVID-19 survey was hosted by Qualtrics and was only available in English. The survey was optimized for both mobile devices and desktop computers. Participants completed the screener, consent, and COVID-19 survey. All data were collected via a self-administered survey without an audio component. The survey took between 10 and 15 min to complete.

### 2.4. Measures

Age: Participants were asked to report their age. The responses were then recoded into the following groups: 18–29, 30–39, 40–49, and 50 and older.

Race and Ethnicity: Racial and ethnic background data were reported by participants using the following response categories: Asian, Black/African American, Hispanic or Latinx, Middle Eastern/North African, Native American or Alaska Native, Pacific Islander, White, or a different identity with the option to specify. The responses were then collapsed into the following groups: White Non-Hispanic, Black Non-Hispanic, Latinx, Asian/Native American/Pacific Islander, and Multi-racial Non-Hispanic.

Gender Identity. All participants were asked to report both their sex assigned at birth and their gender identity. Sex assigned at birth was recorded via a binary response choice: male or female. For gender identity, participants could select from the following categories: male, female, transgender male, transgender female, non-binary/genderqueer/gender non-conforming, or different identity with the option to specify. Participants who reported their gender as male and their sex assigned at birth as male were then recoded as cisgender males. The same process was used to recode for cisgender women (participants who reported their gender as female and indicated their sex at birth was female). Participants who selected male as their gender and female for sex assigned at birth were recoded as transgender men and those who selected female as their gender and male for sex assigned at birth were recoded as transgender women. Those who reported their gender as transgender women or transgender men remained in those categories. Participants who selected non-binary/genderqueer/gender non-conforming remained in this category. For the purposes of this analysis, all transgender and non-binary/genderqueer/gender non-conforming participants were recoded into one group.

Sexual Orientation: Participants were asked to report their sexual orientation using the following response choices: heterosexual or straight, gay or lesbian, bisexual, or a different identity with the option to specify. Heterosexual participants were removed from the sample unless they reported a transgender or non-binary/genderqueer/gender non-conforming gender identity. For the purposes of this analysis, the following groups were created: heterosexual, gay or lesbian, bisexual or pansexual, asexual, queer, and other sexual orientation.

Relationship Status: Participants were asked about their relationship status and were given the following response choices: single, in a committed relationship (without legal status), in a domestic partnership/civil union/other legal status besides marriage, married, separated from partner/divorced, widowed. For the purposes of this analysis, the following groups were created: single, married/domestic partnership, in a committed relationship, and separated/divorced/widowed.

Employment Status: Participants could report their employment status using the following response choices: employed full-time, employed part-time, and unemployed.

Insurance Status: Participants were asked to provide information on their insurance status. They could choose from the following response choices: having private insurance, having public insurance (i.e., Medicaid or Medicare), and being uninsured.

Lifetime Use of Mental Health Medications: Participants were asked about their lifetime use of mental health medications via the following question: "Before 13 March 2020, were you taking any medications for a diagnosed mental health condition (e.g., anxiety, depression, psychosis)?". Responses were collected dichotomously (yes/no).

Access to Mental Health Medications: Participants were asked about their access to mental health medications during the COVID-19 pandemic via the following question: "Since 13 March 2020 have you had trouble getting your psychiatric medications?". Responses were collected via the following response categories: yes, no, and I have not tried to get psychiatric medications since 13 March.

History of Substance Use: Participants were asked if they had used any of the following substances in the past 12 months: cocaine (powder), crack cocaine, ecstasy (also known as molly, MDMA), gamma hydroxybutyrate (GHB), opioids (such as heroin, fentanyl, ketamine, marijuana, morphine), methamphetamine (including crystal meth), poppers, inhalants other than poppers, and hallucinogens (such as LSD, mushrooms). Any use of one of these substances was coded as having a history of substance use, and participants who did not report the use of any of these substances were coded as not having a history of substance use.

Depression Severity: Participants were asked to complete the PHQ scale. This scale contains nine items that assess depressive symptoms in the last two weeks. Items in this scale were summed to create total scores that correspond to depression severity ratings [43]. For the purposes of this study, the responses were dichotomized: minimal to moderate

depressive symptoms (i.e., total scores of 1–14) and moderately severe to severe depressive symptoms (i.e., total scores of 15–27).

*2.5. Analytic Plan*

Descriptive statistics were computed for all predictor variables (age, race and ethnicity, gender identity, sexual orientation, relationship status, employment status, insurance status, lifetime use of mental health medications, access to mental health medications, experiences of intimate partner violence, and a history of substance use) and the outcome variable of depression as assessed via the PHQ-9. All variables were collapsed as previously described prior to the bivariate analysis. The multivariable analysis used multivariable logistic regression to test the relationships between socio-demographic characteristics and depression as assessed via the PHQ-9. All analyses were conducted using SPSS version 23 (IBM Corporation, Armonk, NY, USA).

## 3. Results

*3.1. Sample Characteristics*

Table 1 describes the sociodemographic characteristics of the 1090 participants who met the inclusion criteria.

**Table 1.** Sociodemographic data for a sample of LGBTQ+ individuals during the COVID-19 pandemic, *n* = 1090, 2020, United States.

|  | **Full Sample** <br> ***n* = 1090** <br> **% (*n*)** |
| --- | --- |
| Age | (*M* 33.9, *SD* 11.9, *Range* 18–81) |
| 18–29 | 45.6 (497) |
| 30–39 | 31.8 (347) |
| 40–49 | 10.1 (110) |
| 50 and older | 12.5 (136) |
| Race/Ethnicity | |
| White Non-Hispanic | 69.0 (752) |
| Black Non-Hispanic | 11.1 (121) |
| Latinx | 9.5 (104) |
| Asian, Pacific Islander, or Native American Non-Hispanic | 6.4 (70) |
| Multi-racial Non-Hispanic | 3.9 (43) |
| Gender Identity | |
| Cisgender Men | 44.6 (486) |
| Cisgender Women | 37.6 (410) |
| Transgender Men | 4.0 (44) |
| Transgender Women | 2.6 (28) |
| Non-Binary, Genderqueer, or Gender Non-Conforming | 11.2 (112) |
| Sexual Orientation | |
| Heterosexual | 1.1 (12) |
| Gay or Lesbian | 59.4 (647) |
| Bisexual or Pansexual | 29.5 (322) |
| Asexual | 2.0 (22) |
| Queer | 5.7 (62) |
| Other Sexual Orientation | 2.3 (25) |

**Table 1.** *Cont.*

|  | Full Sample<br>*n* = 1090<br>% (*n*) |
|---|---|
| **Relationship Status** | |
| Single | 29.8 (325) |
| In a Committed Relationship | 39.8 (434) |
| Married/Domestic Partnership | 27.2 (296) |
| Separated/Divorced/Widowed | 3.2 (35) |
| **Employment Status** | |
| Employed Full-Time | 19.7 (215) |
| Employed Part-Time | 59.0 (643) |
| Unemployed | 21.3 (232) |
| **Insurance Status** | |
| Private Insurance | 68.4 (746) |
| Public Insurance | 25.8 (281) |
| Uninsured | 5.8 (63) |
| **Lifetime Use of Mental Health Medications** | |
| Yes | 31.6 (344) |
| No | 68.3 (744) |
| Missing | 0.2 (2) |
| **Trouble Accessing Mental Health Medications** | |
| Yes | 7.2 (78) |
| No | 22.0 (240) |
| Have Not Tried to Access | 2.3 (25) |
| Do Not Take Mental Health Meds | 68.3 (744) |
| Missing | 0.3 (3) |
| **History of Substance Use** | |
| Yes | 51.6 (562) |
| No | 48.4 (528) |
| **Depression Severity** | |
| Minimal to Moderate Depressive Symptoms | 76.2 (831) |
| Moderately Severe to Severe Depressive Symptoms | 23.6 (257) |
| Missing | 0.2 (2) |

Independent variables. The sample in this study mostly comprised participants between the ages of 18 and 29 (45.6%, *n* = 497, SD 11.85, M 33.88). The sample was primarily White Non-Hispanic (69.0%, *n* = 752), with a similar proportion of participants reporting their racial and ethnic background as Black Non-Hispanic (11.1%, *n* = 121) and Latinx (9.5% *n* = 104). The overwhelming majority of the sample were cisgender individuals (men: 44.6%, *n* = 486, women: 37.6%, *n* = 410). Of note, 11.2% (*n* = 122) of participants identified as non-binary, genderqueer, or gender non-conforming. Gay and lesbian participants were the majority in our sample (59.4%, *n* = 647), with nearly one third of the participants reporting their sexual orientation as bisexual or pansexual (29.5%, *n* = 322). For our sample, there was an approximately even distribution between participants who were single (29.8%, *n* = 325), those who were married or in a domestic partnership (27.2%, *n* = 296), and those in a committed relationship without a legal status (39.8%, *n* = 434). Nearly two thirds of participants were employed full-time (59.0%, *n* = 643). Approximately seven in ten participants had private health insurance (68.4%, *n* = 746). Slightly over two thirds of our sample reported using medication for mental health concerns (68.3%, *n* = 744). Nearly one in four participants did not have trouble accessing mental health medications (22.0%, *n* = 240). Finally, slightly over half of the sample had a history of substance use (51.65, *n* = 562).

Dependent Variable. Using the PHQ-9 validated scale [43], slightly less than four in five participants had minimal to moderate depressive symptoms (76.2%, *n* = 831) as compared to 23.6% (*n* = 257) who reported moderately severe to severe depressive symptoms.

### 3.2. Bivariate Analysis

Table 2 presents the results of chi-square tests examining depression severity among members of the LGBTQ+ community. Experiencing moderately severe to severe depressive symptoms was related to age ($\chi^2$ = 27.29; df = 3, $p$ < 0.001); 56.8% (*n* = 146) of participants between the ages of 18 and 29 reported moderately severe to severe depressive symptoms as compared to 31.5% (*n* = 81) of participants between the ages of 30 and 39, 6.2% (*n* = 12) of participants between the ages of 40 and 49, and 5.4% (*n* = 14) of participants aged 50 and older. Gender identity was a significant factor for experiencing moderately severe to severe depressive symptoms ($\chi^2$ = 9.52; df = 4, $p$ = 0.049). Cisgender men reported experiencing more severe depressive symptoms (43.6%, *n* = 112) as compared to cisgender women (22.15, *n* = 85), transgender men (5.1% *n* = 13), transgender women (2.3%, *n* = 6), and non-binary, genderqueer, and gender non-conforming individuals (16.0%, *n* = 41). Similarly, experiencing moderately severe to severe depressive symptoms was related to sexual orientation ($\chi^2$ = 17.03; df = 5, $p$ = 0.004); 51.4% (*n* = 132) of gay and lesbian participants reported moderately severe to severe depressive symptoms as compared to 38.9% (*n* = 100) of bisexual and pansexual participants, 5.8% (*n* = 15) of queer participants, 2.3 (*n* = 6) of asexual participants, 1.2% (*n* = 3) of participants of other sexual orientations, and 0.4% (*n* = 1) of heterosexual participants who reported moderately severe to severe depressive symptoms.

**Table 2.** Depression severity among a sample of LGBTQ+ individuals during the COVID-19 pandemic, *n* = 1090, 2020, United States.

| | Minimal to Moderate Depressive Symptoms *n* = 831 % (*n*) | Moderately Severe to Severe Depressive Symptoms *n* = 257 % (*n*) | $\chi^2$ (df, *p*-Value) |
|---|---|---|---|
| Age | | | 27.293 *** (3, <0.001) |
| 18–29 | 42.2 (351) | 56.8 (146) | - |
| 30–39 | 31.9 (265) | 31.5 (81) | - |
| 40–49 | 11.3 (94) | 6.2 (16) | - |
| 50 and older | 14.6 (121) | 5.4 (14) | - |
| Race/Ethnicity | | | 3.153 (4, 0.533) |
| White Non-Hispanic | 69.2 (575) | 68.1 (175) | - |
| Black Non-Hispanic | 10.6 (88) | 12.8 (33) | - |
| Latinx | 10.2 (85) | 7.4 (19) | - |
| Asian, Pacific Islander, or Native American Non-Hispanic | 6.1 (51) | 7.4 (19) | - |
| Multi-racial Non-Hispanic | 3.9 (32) | 4.3 (11) | - |
| Gender Identity | | | 9.520 * (4, 0.049) |
| Cisgender Men | 45.0 (374) | 43.6 (112) | - |
| Cisgender Women | 38.9 (323) | 33.1 (85) | - |
| Transgender Men | 3.7 (31) | 5.1 (12) | - |
| Transgender Women | 2.6 (22) | 2.3 (6) | - |
| Non-Binary, Genderqueer, or Gender Non-Conforming | 9.7 (81) | 16.0 (41) | - |

**Table 2.** *Cont.*

| | Minimal to Moderate Depressive Symptoms *n* = 831 % (*n*) | Moderately Severe to Severe Depressive Symptoms *n* = 257 % (*n*) | $\chi^2$ (df, *p*-Value) |
|---|---|---|---|
| Sexual Orientation | | | 17.026 ** (5, 0.004) |
| Heterosexual | 1.3 (11) | 0.4 (1) | - |
| Gay or Lesbian | 61.7 (513) | 51.4 (132) | - |
| Bisexual or Pansexual | 26.7 (222) | 38.9 (100) | - |
| Asexual | 1.9 (16) | 2.3 (6) | - |
| Queer | 5.7 (47) | 5.8 (15) | - |
| Other Sexual Orientation | 2.6 (22) | 1.2 (3) | - |
| Relationship Status | | | 7.749 (3, 0.051) |
| Single | 29.1 (242) | 32.3 (83) | - |
| In a Committed Relationship | 38.5 (320) | 44.4 (114) | - |
| Married/Domestic Partnership | 29.1 (242) | 20.6 (53) | - |
| Separated/Divorced/Widowed | 3.2 (27) | 2.7 (7) | - |
| Employment Status | | | 58.951 *** (2, <0.001) |
| Employed Full-Time | 65.2 (542) | 39.3 (101) | - |
| Employed Part-Time | 16.8 (140) | 35.0 (90) | - |
| Unemployed | 17.9 (149) | 25.7 (66) | - |
| Insurance Status | | | 16.423 *** (2, <0.001) |
| Private Insurance | 71.7 (596) | 58.4 (150) | - |
| Public Insurance | 23.2 (193) | 33.5 (86) | - |
| Uninsured | 5.1 (42) | 8.2 (21) | - |
| Lifetime Use of Mental Health Medications | | | 2.719 (1, 0.099) |
| Yes | 30.3 (252) | 35.8 (92) | - |
| No | 69.7 (579) | 64.2 (165) | - |
| Trouble Accessing Mental Health Medications | | | 3.693 (2, 0.0158) |
| Yes | 21.5 (54) | 26.1 (24) | - |
| No | 72.5 (182) | 63.0 (58) | - |
| Have Not Tried to Access | 6.0 (15) | 10.9 (10) | - |
| History of Substance Use | | | 9.443 ** (1, 0.002) |
| Yes | 45.8 (381) | 56.8 (146) | - |
| No | 54.2 (450) | 43.2 (111) | - |

Note: * $p < 0.05$, ** $p < 0.01$, *** $p < 0.001$.

Whether a survey participant reported experiencing moderately severe to severe depressive symptoms was related to employment status ($\chi^2$ = 58.95; df = 2; $p < 0.001$); 39.3% (*n* = 101) of participants employed full-time reported moderately severe to severe depressive symptoms, whereas only 35.0% (*n* = 90) of part-time-employed participants and 25.7% (*n* = 66) of unemployed participants reported moderately severe to severe depressive symptoms. Insurance status was also significantly related to experiencing moderately severe to severe depressive symptoms ($\chi^2$ = 16.42; df = 2; $p < 0.001$). In total, 58.4% (*n* = 150) of participants with private insurance reported moderately severe to severe depressive symptoms, whereas 33.5% (*n* = 86) of participants with public insurance and 8.2% (*n* = 21) of uninsured participants reported experiencing moderately severe to severe depressive symptoms. Finally, experiencing moderately severe to severe depressive symptoms was related to participants' history of substance use ($\chi^2$ = 9.44; df = 1; $p = 0.002$); 58.6% (*n* = 146) of participants with a history of substance use experienced moderately severe to severe depressive symptoms as compared to only 43.2% (*n* = 111) of participants without a history of substance use.

### 3.3. Multivariable Analysis

Table 3 presents the results of the multivariable logistic regression models for depression severity among participants in our sample. The adjusted model for depression severity was significant ($\chi^2$ = 92.744; df = 15; $p$ < 0.001) with Nagelkerke R2 = 12.3%. Participants aged 50 years old or older were more likely to report minimal to moderately severe depressive symptoms as compared to all other age groups (AOR = 0.333, 95% CI 0.177–0.626, $p$ < 0.001). Participants who were unemployed were less likely to report moderately severe to severe depressive symptoms as compared to groups who were employed full-time (AOR = 0.474, 95% CI 0.321–0.701, $p$ < 0.001). Finally, people with a history of substance use were less likely to report moderately severe to severe depressive symptoms as compared to participants without a history of substance use (AOR = 0.680, 95% CI 0.505–0.915, $p$ = 0.011).

**Table 3.** Multivariable logistic regression models examining depression severity among a sample of LGBTQ+ individuals during the COVID-19 pandemic, $n$ = 1090, 2020, United States.

| | Unadjusted Model OR, 95% CI, *p*-Value | Adjusted Model OR, 95% CI, *p*-Value |
|---|---|---|
| Age | | |
| 18–29 [†] | - | |
| 30–39 | 0.735, [0.536–1.01], 0.055 | 0.917, [0.652–1.290], 0.618 |
| 40–49 | 0.409 **, [0.233–0.719], 0.002 | 0.595, [0.326–1.085], 0.090 |
| 50 and older | 0.278 ***, [0.155–0.500], <0.001 | 0.333 ***, [0.177–0.626], <0.001 |
| Race/Ethnicity | | |
| White Non-Hispanic [†] | - | - |
| Black Non-Hispanic | 1.007, [0.519–1.950], 0.984 | - |
| Latinx | 0.817, [0.470–1.421], 0.474 | - |
| Asian, Pacific Islander, or Native American Non-Hispanic | 0.600, [0.291–1.238], 0.167 | - |
| Multi-racial Non-Hispanic | 0.923, [0.389–2.190], 0.855 | - |
| Gender Identity | | |
| Cisgender Men [†] | - | - |
| Cisgender Women | 0.879, [0.639–1.209], 0.427 | 0.767, [0.545–1.079], 0.127 |
| Transgender Men | 1.400, [0.709–2.767], 0.333 | 1.052, [0.511–2.166], 0.890 |
| Transgender Women | 0.911, [0.360–2.301], 0.843 | 0.739, [0.283–1.929], 0.537 |
| Non-Binary, Genderqueer, or Gender Non-Conforming | 1.690 *, [1.099–2.601], 0.017 | 1.356, [0.855–2.149], 0.196 |
| Sexual Orientation | | |
| Heterosexual [†] | - | - |
| Gay or Lesbian | 2.830, [0.362–22.11], 0.321 | - |
| Bisexual or Pansexual | 4.955, [0.631–38.903], 0.128 | - |
| Queer | 3.511, [0.418–29.483], 0.247 | - |
| Asexual | 4.125, [0.434–39.211], 0.217 | - |
| Other Sexual Orientation | 1.500, [0.139–16.144], 0.738 | - |
| Relationship Status | | |
| Single [†] | - | - |
| In a Committed Relationship | 1.039, [0.748–1.443], 0.821 | 0.942, [0.663–1.338], 0.739 |
| Married/Domestic Partnership | 0.639 *, [0.433–0.941], 0.023 | 0.903, [0.593–1.376], 0.636 |
| Separated/Divorced/Widowed | 0.756, [0.317–1.800], 0.527 | 1.219, [0.481–3.085], 0.677 |
| Employment Status | | |
| Unemployed [†] | - | - |
| Employed Full-Time | 0.421 ***, [0.294–0.603], <0.001 | 0.474 ***, [0.321–0.701], <0.001 |
| Employed Part-Time | 1.451, [0.980–2.145], 0.063 | 1.342, [0.891, 2.021], 0.159 |
| Insurance Status | | |
| Uninsured [†] | - | - |
| Private Insurance | 0.503*, [0.289–0.876], 0.015 | 0.779, [0.422–1.403], 0.405 |
| Public Insurance | 0.891, [0.498–1.595], 0.698 | 1.031, [0.561–1.897], 0.921 |

**Table 3.** *Cont.*

|  | Unadjusted Model OR, 95% CI, *p*-Value | Adjusted Model OR, 95% CI, *p*-Value |
|---|---|---|
| Lifetime Use of Mental Health Medications |  |  |
| No [†] | - | - |
| Yes | 0.781, [0.581–1.048], 0.100 | - |
| Trouble Accessing Mental Health Medications |  |  |
| No [†] | - | - |
| Yes | 0.478, [0.204–1.122], 0.090 | - |
| Have Not Tried to Access | 0.667, [0.262–1.696], 0.395 | - |
| History of Substance Use |  |  |
| No [†] | - | - |
| Yes | 0.644 **, [0.486–0.853], 0.002 | 0.680 *, [0.505–0.915], 0.011 |

Note: * $p < 0.05$, ** $p < 0.01$, *** $p < 0.001$. [†] indicates reference group.

## 4. Discussion

Although the U.S. has largely removed prior COVID-19 policy measures as the virus is now considered endemic [44] and public concern around COVID is extremely low [45], it is critical to understand how people coped during this fraught time period. Throughout the height of the pandemic, researchers and community leaders were able to identify a disproportionate burden on minoritized communities in real time [46–48]. Salerno et al. (2020) outline the social-, structural-, and individual-level factors that placed LGBTQ people at increased risk for worse mental health outcomes, including pervasive structural vulnerability, higher rates of poverty, and increased social isolation due to lack of born family support [20]. The importance of social support was rendered particularly visible during the height of the pandemic. According to the findings of the present study, participants aged 50 years old or older were more likely to report minimal to moderately severe depressive symptoms compared to all other age groups. This aligns with previous literature that has found that older LGBTQ individuals were more likely to report emotional distress than non-LGBTQ people during the pandemic [49]. Increased attention to the needs of older LGBTQ individuals, especially during times of national and global emergency, is warranted, as this group is generally more vulnerable, having experienced decades of stigma, criminalization, discrimination, grief, victimization, and social and economic exclusion [50].

Unemployment and under-employment issues were exacerbated during the height of the pandemic. This analysis found that participants who were unemployed were less likely to report moderately severe to severe depressive symptoms as compared to groups who were employed full-time. This finding could be interpreted in several ways. First, individuals who were unemployed may have had a lower exposure to COVID-19, serving as a protective factor. Additionally, Amerikaner and colleagues found that LGBTQ adults working from home reported lower stress and tiredness compared to those working in their workplace [51]. Related to this, working in the workplace had a more negative effect on LGBTQ adults than non-LGBTQ adults working in the workplace, suggesting that not being in a workplace setting perhaps reduced "minority stress" [51]. This study aligns with our finding that unemployed individuals, i.e., those not working in a workplace, reported lower depressive symptoms. Although much of the COVID 19-related literature expands upon the detrimental effects of unemployment, of which there are many [52,53], the findings of this analysis suggest that there were protective factors at play that correlated with lower depressive symptoms, such as reduced exposure to harmful discrimination and stigma [51].

The extant literature widely documents the increased use of substances during the COVID-19 pandemic [25,54–57]. This study found that people with a history of substance use were less likely to report moderately severe to severe depressive symptoms as compared

to participants without a history of substance use. This finding suggests that people were likely relying on cannabis, alcohol, and other drugs to cope with worsened mental health during the pandemic. Although it does not necessarily indicate a causal relationship, this finding may suggest that not all substance use is inherently problematic. More broadly, this likely reliance on substances to cope highlights the problems with the U.S. healthcare system. Given that health insurance is largely linked to employment in the U.S., many individuals lost coverage during this period [54]. Even for those who retained health insurance, mental healthcare in the U.S. is notoriously lacking [55]. Indeed, the sharp increase in need for mental health professionals, against the backdrop of pre-existing mental healthcare provider shortages across many regions of the U.S., resulted in exacerbated shortages for individuals seeking support during the height of the pandemic. This is especially concerning for LGBTQ individuals, as most health providers are not trained to support sexual- and gender-minority people [25,57], though effective interventions exist [56].

Although the study presented here found that those with a substance use history were less likely to report severe depressive symptoms, other studies have found different results. In a study of pandemic cannabis use and mental health, Gattamorta and colleagues (2021) found that LGBTQ users reported higher levels of alcohol use and more symptoms of depression and anxiety than non-LGBTQ people who use cannabis [36]. Those LBGTQ individuals who had higher levels of mental health distress also reported having COVID infection worries, concerns about loved ones, and worries about changes in their sexual activity [36]. Although the authors documented increased problematic use, both this study and Gattamorta et al.'s study provide support for increased substance use and mental health-related services [36]. People who use drugs deserve to be treated with dignity and have access to harm reduction tools [58]. Harm reduction efforts have been firmly established as saving lives [59], saving taxpayers' money [60], and reducing the burden on healthcare systems and first responders [61]. During the pandemic, overdose deaths surpassed 100,000 people [62], highlighting the urgent need for increased efforts around addressing stigma, harm reduction, substance use treatment availability (particularly medications for opioid use disorder), and public health approaches to drug use rather than criminal ones [63].

## 5. Implications

Understanding the long-lasting impacts of the COVID-19 pandemic is essential for allocating public health funding towards mental health services, as well as developing policies and programs that will improve access to mental health services. Health policy changed after President Biden declared the end of the national emergency funds for COVID-19. The most significant change was the end of insurance reimbursement for telemedicine office visits for health services, including mental health services. Providing mental health services via telemedicine increases access to these services to meet the needs of patients. This study serves to provide an estimate of depression and depression severity among LGBTQ+ individuals in the United States during the COVID-19 pandemic. Further research is needed to examine the change in depression onset and changes in depression severity among this population as the COVID-19 pandemic continues well beyond the initial outbreak at the beginning of 2020. This study serves as an important first step in establishing the prevalence of depression among LGBTQ+ individuals in the United States.

## 6. Limitations

This study is not without limitations. First, this study employed an internet-based recruitment strategy due to the ongoing COVID-19 pandemic. While data cleaning protocols were used, it is possible that there are duplicate responses and that bot-based responses were not detected by our data cleaning protocol. Second, our sample comprised a majority of white participants, despite efforts to recruit participants of color. This limits the generalizability of our findings. Third, this is a cross-sectional study, and the study design does not

allow for causal inferences. Fourth, although this study used a validated tool for assessing depression symptomology, social desirability bias likely resulted in the under-reporting of depression-related behaviors and feelings.

## 7. Conclusions

This study demonstrated that the COVID-19 pandemic had wide-ranging implications for the mental health of LGBTQ+ individuals in the United States. As a cross-sectional study, this manuscript is only able to offer a glimpse into a particular moment in the pandemic; yet the extant literature on mental health provides evidence that mental health issues do not resolve after a stressor is removed. Furthermore, stressors that were brought to light during the pandemic, specifically the death of loved ones, loss of employment, and concerns around health, have implications for mental health outcomes that persist across an individual's lifespan.

**Author Contributions:** Conceptualization, M.G., J.J., T.O., J.G. and C.B.S.; methodology, M.G., J.G. and C.B.S.; software, M.G.; validation, M.G.; formal analysis, M.G. and J.G.; investigation, M.G., J.J., T.O. and J.G.; resources, M.G.; data curation, M.G. and J.G.; writing—original draft preparation, M.G., J.J., T.O. and J.G.; writing—review and editing, M.G. and P.N.H.; visualization, M.G. and J.G.; supervision, M.G. and P.N.H.; project administration, P.N.H.; funding acquisition, P.N.H. All authors have read and agreed to the published version of the manuscript.

**Funding:** This research received no external funding.

**Institutional Review Board Statement:** The study was conducted according to the guidelines of the Declaration of Helsinki, and approved by the Institutional Review Board of Rutgers University #Pro2020000920 approved March 2020.

**Informed Consent Statement:** Informed consent was obtained from all subjects involved in this study.

**Data Availability Statement:** The data presented in this study are available on request from the corresponding author.

**Acknowledgments:** The authors would like to acknowledge the following researchers for their initial contributions to this manuscript; Rafael Perez-Figueroa and Anita Karr.

**Conflicts of Interest:** The authors declare no conflict of interest.

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
