# Peer review of "Depression Severity among a Sample of LGBTQ+ Individuals during the COVID-19 Pandemic"

_societies, doi:10.3390/soc13110244_

Round 1

Reviewer 1 Report

Comments and Suggestions for Authors

You have a really interesting data set which is well analysed with good explanation of the findings.

However, the introduction needs developing. For example in Line 51-52 what sort of discrimination are you referring to. The reader needs to be able to understand this without going to the source material. Also how does substance use differ from other populations.

Similarly the discussion needs developing. Firstly you make statements that go well beyond the data, please tighten these up. Also, you could discuss areas where you thought that there would be significance but there wasn't.
Finally it is important to acknowledge the limitations of the study.

Note. Lines 50-57 need checking for sense. They are hard to understand.

Author Response

COMMENTS FROM REVIEWER 1

You have a really interesting data set which is well analysed with good explanation of the findings. However, the introduction needs developing. For example in Line 51-52 what sort of discrimination are you referring to. The reader needs to be able to understand this without going to the source material. Also how does substance use differ from other populations.

RESPONSE TO REVIEWER: Thank you for this comment.  We have revised the entire introduction.  The paragraph referenced in this comment now read as follows: Higher levels of stress and depression during the pandemic may be explained by experiences of discrimination related to sexual orientation [6, 32-34] as well as an increase in frequency of intimate partner violence [6, 35]. Furthermore, social isolation, loneliness, and lack of access to community groups may have more severely impacted the mental health of LGBTQ+ people due to this populations’ tendency to rely more on chosen family for support [31]. For example, LGBTQ+ young people who were forced to cohabitate with unsupportive family members during stay-at-home orders experienced heightened psychological distress [36, 37]

Similarly the discussion needs developing. Firstly you make statements that go well beyond the data, please tighten these up. Also, you could discuss areas where you thought that there would be significance but there wasn't.

RESPONSE TO REVIEWER: Thank you for this comment. We have made revisions to the discussion to ensure that all comments are directly related to the data employed by this analysis.

Finally it is important to acknowledge the limitations of the study.

RESPONSE TO REVIEWER: Thank you for this comment.  We apologize for the oversight and have included a limitations paragraph that reads as follows: This study is not without limitations.  First, this study employed an internet based recruitment strategy due to the ongoing COVID-19 pandemic.  While data cleaning protocols were used it is possible that there are duplicate responses and that bot-based responses were not detected by our data cleaning protocol. Second, our sample is comprised of a majority of white participants despite efforts to recruit participants of color.  This limits the generalizability of our findings. Third, this is a cross-sectional study and the study design does not allow for causal inferences. Fourth, although this study used a validated tool for assessing depression symptomology, social desirability bias likely resulted in the under reporting of depression-related behaviors and feelings.

Note. Lines 50-57 need checking for sense. They are hard to understand.

RESPONSE TO REVIEWER: Thank you for this comment. We have revised this paragraph to read as follows:  Higher levels of stress and depression during the pandemic may be explained by experiences of discrimination related to sexual orientation [6, 32-34] as well as an increase in frequency of intimate partner violence [6, 35]. Furthermore, social isolation, loneliness, and lack of access to community groups may have more severely impacted the mental health of LGBTQ+ people due to this populations’ tendency to rely more on chosen family for support [31]. For example, LGBTQ+ young people who were forced to cohabitate with unsupportive family members during stay-at-home orders experienced heightened psychological distress [36, 37]

Reviewer 2 Report

Comments and Suggestions for Authors

A very interesting paper, referred to an actual topic as the effects of COVID about the Depression Severity among a Sample of LGBTQ+ Individuals 2 during the  Pandemic.  I would like to make some comments and suggestions for the
authors, in order to help for the publication:

The abstract needs a review,  the objective of the article  ("This analysis examines how the COVID-19 pandemic has amplified depressive symptoms occurring in an LGBTQ+ sample (n=1090) in the United States. Variables examined included socio-demographic factors, access to healthcare, use of mental health medication, and experiences of interpersonal violence among others") has got implicit the opinion of the authors, which is a lack of scientific sound. I highly recommend using the structure Background-Methods-Results-Conclusions.
- In Keywords, there is no mention to the methods or aims of the study.
- In my opinion, the Introduction is very well done.
- The Methods part is good in content, the use of methodology is a strength of the study but there is a lack of explanation: how the authors have identified the increase/decrease of Depression Severity (or other mental health problems) in the study is not explained in the text, and it's fundamental for this paper the explanation.
- The description/identification of possible bias or limitations is not included in the paper, and it's also a fundamental part of research.

- Results are wonderful in content. 
- Discussion needs a review, especially because there is no part of Biases or Limitations. These ones must add all the possible limitations of the study, especially about methods and analysis. I think it’s fundamental for its publication. Also, as I said previously, there is no mention about the biases in the text, and they are really clear when you read the paper. Please, include the Limitations and biases.
- Also, the part of Conclusions doesn't have practical proposals, and in my opinion, this is a paper with a clear practical motivation to decrease the psychosocial impact of lockdowns (for LGTBIQ+ populations or other groups)

Author Response

Reviewer 2:
A very interesting paper, referred to an actual topic as the effects of COVID about the Depression Severity among a Sample of LGBTQ+ Individuals 2 during the  Pandemic.  I would like to make some comments and suggestions for the authors, in order to help for the publication:

The abstract needs a review,  the objective of the article  ("This analysis examines how the COVID-19 pandemic has amplified depressive symptoms occurring in an LGBTQ+ sample (n=1090) in the United States. Variables examined included socio-demographic factors, access to healthcare, use of mental health medication, and experiences of interpersonal violence among others") has got implicit the opinion of the authors, which is a lack of scientific sound. I highly recommend using the structure Background-Methods-Results-Conclusions.

RESPONSE TO REVIEWER: Thank you for this comment.  We have removed the implicit opinion and formatted the abstract as a structured abstract.

Background: The global pandemic of coronavirus disease 2019 (COVID-19) has wrought immense impacts to global community health, public perception of healthcare, and attitudes surrounding mental health during widespread quarantine. This

Methods: This analysis examines the rates of depressive symptomology among a sample of LGBTQ+ identified individuals in the United States (n=1090). Variables examined included socio-demographic factors, access to healthcare, use of mental health medication, and experiences of interpersonal violence among others. 

Results: Findings indicate depressive symptoms were more severe for older adults who tend not to seek services due to mental health stigma; whereas those who are were not working and those who were using substances were less likely to report depressive symptoms.  Those who were unemployed may have reported less severe depressive symptoms as staying at home reduced their exposure to COVID-19.  Similarly, substance use may have acted as a means of coping with depression and anxiety related to the pandemic. 

Conclusions: Understanding the mental health of marginalized populations such as the LGBTQ+ community is critical to provide more nuanced preventative health care for the unique populations as members of the LGBTQ+ community are non-monolithic and require more personalized approaches to their health care needs.

- In Keywords, there is no mention to the methods or aims of the study.

RESPONSE TO REVIEWER: Thank you for this comment. We have revised the keywords to read: Key words: COVID-19, LGBTQ+, depression, employment loss, substance use, multivariable logistic regression, pandemic impact on mental health

- In my opinion, the Introduction is very well done.

RESPONSE TO REVIEWER: Thank you very much for this feedback.  We had a student/early career professional work on this section and this praise will be passed on to them and have a special significance to them.

- The Methods part is good in content, the use of methodology is a strength of the study but there is a lack of explanation: how the authors have identified the increase/decrease of Depression Severity (or other mental health problems) in the study is not explained in the text, and it's fundamental for this paper the explanation.

RESPONSE TO REVIEWER: Thank you for this comment.  It is not our intention to study the increase/decrease in depression severity among LGBTQ+ individuals; our intention was merely to provide estimates of depression and the severity (minimal to moderate and moderately severe to severe).  Since this is a cross-sectional study, we did not conduct repeated measures, merely point in time estimates. We have removed all implications of assessing a change in depression severity from the manuscript.

- The description/identification of possible bias or limitations is not included in the paper, and it's also a fundamental part of research.

RESPONSE TO REVIEWER: Thank you for this comment.  We apologize for the oversight and have included a limitations paragraph that reads as follows: This study is not without limitations.  First, this study employed an internet based recruitment strategy due to the ongoing COVID-19 pandemic.  While data cleaning protocols were used it is possible that there are duplicate responses and that bot-based responses were not detected by our data cleaning protocol. Second, our sample is comprised of a majority of white participants despite efforts to recruit participants of color.  This limits the generalizability of our findings. Third, this is a cross-sectional study and the study design does not allow for causal inferences. Fourth, although this study used a validated tool for assessing depression symptomology, social desirability bias likely resulted in the under reporting of depression-related behaviors and feelings.

- Results are wonderful in content. 

RESPONSE TO REVIEWER: Thank you again for this praise, encouragement in academia is rare and much appreciated.

- Discussion needs a review, especially because there is no part of Biases or Limitations. These ones must add all the possible limitations of the study, especially about methods and analysis. I think it’s fundamental for its publication. Also, as I said previously, there is no mention about the biases in the text, and they are really clear when you read the paper. Please, include the Limitations and biases.

RESPONSE TO REVIEWER: Thank you for this comment.  We apologize for the oversight and have included a limitations paragraph that reads as follows: This study is not without limitations.  First, this study employed an internet based recruitment strategy due to the ongoing COVID-19 pandemic.  While data cleaning protocols were used it is possible that there are duplicate responses and that bot-based responses were not detected by our data cleaning protocol. Second, our sample is comprised of a majority of white participants despite efforts to recruit participants of color.  This limits the generalizability of our findings. Third, this is a cross-sectional study and the study design does not allow for causal inferences. Fourth, although this study used a validated tool for assessing depression symptomology, social desirability bias likely resulted in the under reporting of depression-related behaviors and feelings.

- Also, the part of Conclusions doesn't have practical proposals, and in my opinion, this is a paper with a clear practical motivation to decrease the psychosocial impact of lockdowns (for LGTBIQ+ populations or other groups)

RESPONSE TO REVIEWER: Thank you for this comment. We have added the following paragraph: Understanding the long-lasting impacts of the COVID-19 pandemic is essential for allocating public health funding towards mental health services as well as developing policies and programs that will improve access to mental health services.  Health policy has changed after President Biden declared the end of the national emergency funds for COVID-19.  The most significant change is the end of insurance reimbursement for telemedicine office visits for health services, including mental health services.   Providing mental health services via telemedicine increases access to these services to meet the needs of the patient. This study serves to provide an estimate of depression and depression severity among LGBTQ+ individuals in the United States during the COVID-19 pandemic. Further research is needed to examine the change in depression onset and changes in depression severity among this population as the COVID-19 pandemic continues well beyond the initial outbreak in the beginning of 2020.  This study serves as an important first step in establishing the prevalence of depression among LGBTQ+ individuals in the United States.

Round 2

Reviewer 2 Report

Comments and Suggestions for Authors

Lot of thanks for your efforts for changing the paper.

Author Response

Thank you!